

# Sociodemographic distributions and molecular characterization of colonized *Enterococcus faecium* isolates from locality hospitals in Khartoum, Sudan

Loai Abdelati Siddig[1], Magdi Bayoumi[2] and Nasreldin Elhadi[3]

[1] Microbiology Department, Faculty of Medical Laboratory Sciences, University of Medical Sciences and Technology, Khartoum, Sudan

[2] Microbiology Department, Faculty of Medicine, University of Medical Sciences and Technology (UMST), Khartoum, Sudan

[3] Department of Clinical Laboratory Science, College of Applied Medical Sciences, Imam Abdulrahman Bin Faisal University, Dammam, Kingdom of Saudi Arabia

## ABSTRACT

**Background**. *Enterococcus faecium* is an opportunistic pathogen of humans with diverse hosts, encompassing animals as well as human beings. In the past twenty years, there has been a rise in the instances of nosocomial infections that are linked to antibiotic-resistant *Enterococcus faecium*. The acquisition of diverse antimicrobial resistance factors has driven the global development of robust and convergent adaptive mechanisms within the healthcare environment. The presence of microorganisms in hospitalized and non-hospitalized patient populations has been significantly aided by the facilitation of various perturbations within their respective microbiomes.

**Objective**. This study aimed to determine the antimicrobial profile, demographic and clinical characteristics, along with the detection of virulence encoding genes, and to find out the clonal genetic relationship among colonized *E. faecium* strains.

**Methodology**. A hospital-based cross-sectional study was carried out between October 2018 and March 2020 at four Khartoum locality hospitals in Sudan. The study comprised a total of 108 strains of *E. faecium* isolated from patients admitted to four locality hospitals in Khartoum. A self-structured questionnaire was used to gather information on sociodemographic traits. Data were analyzed using chi-square test. In all cases, $P$ value $\leq 0.05$ with a corresponding 95% confidence interval was considered statistically significant. Moreover, enterobacterial repetitive intergenic consensus–polymerase chain reaction (ERIC-PCR) was utilized to assess the prevalence of clonal relationships, and the gel was analyzed using CLIQS software.

**Results**. In this study, the isolation rate of colonized *E. faecium* strains was 108/170 (63.5%). The colonization of *E. faecium* and its association with various sociodemographic and clinical features was examined. 73 (67.6%) of patients had multidrug-resistant (MDR), and 22 (20.4%) had extensively drug-resistant (XDR), 73 (67.6%) of patients engaged in self-medication practices. Eighty patients (74.1%) were non-adherence to prescribed antibiotics, while 70 (64.8%) patients reported recent antibiotic usage within the 3 months. The present study suggests that demographic factors may not be significantly associated with the incidence of *E. faecium* infection except for patients who had a prior history of antibiotic use ($P \leq 0.005$). The analysis of virulence genes showed a high prevalence of *asa1* gene (22.2%) among strains. In ERIC-PCR

Corresponding author
Loai Abdelati Siddig,
aboallolgah@yahoo.com

the genetic relatedness of *E. faecium* showed seven identical clusters (A–G) with 100% genetic similarity. This implies clonal propagation in hospitals and communities. **Conclusion**. This study found that the incidence of *E. faecium* isolated from locality hospitals in Khartoum was likely due to the spread of *E. faecium* clones, thereby highlighting the need for intensifying infection control measures to prevent the spreading of nosocomial infection.

# INTRODUCTION

In the past two decades, *Enterococcus faecium* has rapidly evolved as a worldwide nosocomial pathogen by successfully adapting to conditions in a nosocomial setting and acquiring resistance against glycopeptides (*Top, Willems & Bonten, 2008*; *Bonten, Willems & Weinstein, 2001*). The nosocomial pathogen *E. faecium* can survive for prolonged periods on surfaces in the absence of nutrients, and also in hospital environments, these traits are thought to contribute to the ability of *E. faecium* to transmit between patients in hospitals (*Freitas et al., 2022*; *Gao, Howden & Stinear, 2018*). It is commonly found in the intestines of humans and animals. While it is generally considered a harmless commensal organism, it sometimes causes infections, especially in individuals with weakened immune systems or other underlying health conditions such as urinary tract infections, bacteremia, and endocarditis. The relative importance of *E. faecium* as a pathogen has increased with the occurrence of high-level resistance to multiple antimicrobial drugs, such as amoxicillin clavulanic acid, aminoglycosides, cephalosporin, and vancomycin (*Siddig et al., 2022*). The prevalence of multidrug-resistant (MDR) *E. faecium* infections is rising globally, although epidemiological research remains generally scarce in specific regions such as African countries including Sudan (*Freitas et al., 2022*). Vancomycin-resistant *E. faecium* (VRE*fm*) is the most common multidrug-resistant *Enterococcus* species and is one of the most serious concerns in healthcare settings (*Lee et al., 2019*; *Murray, 2000*), designated as a high-priority pathogen in need of therapeutic research and development according to the World Health Organization (WHO) and the Centers for Disease Control and Prevention (CDC) (*Tacconelli et al., 2018*; *Centers for Disease Control and Prevention, 2019*). In the invasion process, enterococci use a variety of virulence factors including (*asa*1, *cyl*A, *esp, gel*E, and *hyl* gene) for adhering to the infection site and colonizing (*Mundy, Sahm & Gilmore, 2000*; *Nasaj et al., 2016*), along with the presence of damage to the host tissue and antibiotic resistance, all help with the invasion process. In the end, the clinical manifestation of infection in the targeted vital tissues is the result of interactions between the host and enterococci.

Molecular typing is becoming a paradigm for understanding the fundamental mechanisms of *Enterococcus* infections in hospital settings to investigate the clonal

relationship among bacterial strains, and to trace the source of infections during hospital outbreaks (*Saengsuwan et al., 2021*). PCR-based genotyping tools are used for determining different DNA fingerprints, among several PCR-based tools, the ERIC-PCR discriminating is a rapid, and cost-effective genotyping method for different types of strains. Moreover, there are different numbers of ERIC sequence copies among bacterial species. Interestingly, there is a significant diversity of copy numbers among different strains of *E. faecium* (*Jannati et al., 2020*). The use of homologous primers to ERIC sequences allows for assessing the clonal variability found in various *E. faecium* strains, in which the resultant patterns can be utilized to evaluate the phylogenetic closeness with the help of CLIQS 1D PRO software (*TotalLab, 2015*). In Sudan, in particular, no reports are available in epidemiological studies, assessing antibiotic resistance, molecular analysis, or other features of clinical *E. faecium* isolates.

## MATERIALS & METHODS

### Study design, setting, and period

A hospital-based cross-sectional study was conducted in four tertiary hospitals in the Khartoum locality (The Academy Charity Teaching Hospital (ACTH), Dar-Alelaj Specialized Hospital (DASH), Ibrahim Malik Teaching Hospital (IMTH), and Yastabshiroon Hospital Riyadh (YASH)), from October 2018 to March 2020. Those hospitals provide different levels of care services in all disciplines, four wards were included: Medicine, ICU, Surgical, and Pediatric wards. Ethical approval for this study was obtained from the Graduate College-UMST and the Federal Ministry of Health Reference no. (ws-wk-aete-44-a), Sudan.

### Patient and statistical analysis

Information on patient sociodemographic characteristics, risk factor data, and other independent variables was collected from each participant using a self-structured questionnaire. Data collection was done under the supervision of the project advisors. The information was gathered by conducting in-person interviews with patients in outpatient clinics or hospital wards by attending a nurse from the hospital. Qualitative data were described using numbers and percentages. Comparison between different groups regarding categorical variables was tested using Pearson's chi-square test ($P \leq 0.05$) to investigate the significance of *E. faecium* colonization association with sociodemographic distributions.

### Collection and identification of *Enterococcus faecium*

A total of 108 *E. faecium* were isolated from fecal cultures, from patients in four Khartoum locality hospitals. All specimens were cultivated on the surface of the bile-esculin medium (HiMedia, Maharashtra, India), and incubated at 37 °C for 24 h. Colonies growing on bile-esculin medium with a dark brown halo and colonial morphology resembling enterococci were collected as presumptive isolates. All presumptive isolates of enterococci were further confirmed using phenotype tests such as Gram staining, catalase, oxidase, and growth in 6.5% NaCl broth, as described elsewhere (*Manero & Blanch, 1999*). Catalase test aimed to detect the presence of catalase enzyme in microorganisms. This test was performed by the

addition of two or three drops of 3% $H_2O_2$ onto the cultures. The oxidase test is a valuable technique for the differentiation of bacteria based on their enzymatic capabilities. The test was performed by adding two droplets of oxidase reagent on sterile Petri dishes smeared with a loopful of bacterial growth. Once obtaining pure isolates that met the criteria of being (Gram-positive, and lacking catalase and oxidase activity), additional assays were performed. The isolates also were subjected to incubation in BHI broth that contained 6.5% NaCl at a temperature of 37 °C for a duration of 24 to 48 h. Isolates were presumptively considered to be enterococci based on the characteristic features: Gram-positivity and catalase-and oxidase-negativity, growth in a medium containing 6.5% NaCl.

## Antibiotic susceptibility testing

Utilizing the Kirby-Bauer method's disk diffusion procedure on Muller-Hinton agar (Hi-Media, Maharashtra, India) (*Bauer et al., 1966*), Hundred-eight *E. faecium* strains were investigated. A total of fifteen antibacterial drugs were included and were previously described in *Siddig et al. (2022)*. According to the recommendations of the Hi-Media Antimicrobial Susceptibility Systems, the width of the zone of inhibition was determined and recorded as Sensitive (S), Intermediate (I), or Resistant (R). Antibiotic susceptibility testing was carried out using the reference control organism *E. faecalis* ATCC29212. All of the analyzed strains of *E. faecium* in this study were classified as multidrug-resistant (MDR), extensively drug-resistant (XDR), and pandrug-resistant (PDR). MDR is defined as resistant to three or more antimicrobial classes (*Magiorakos et al., 2012*). Non-susceptibility to at least one agent in all but two or fewer antimicrobial categories is defined as XDR, while PDR is defined as resistance to all agents in all antimicrobial categories.

## DNA extraction and detection of VRE*fm* virulence genes by PCR

Genomic DNA was extracted using the G-spin Genomic DNA Extraction Kit (iNtRON, Seongnam, South Korea) following the manufacturer's instructions. Amplification was performed according to a published protocol (*Vankerckhoven et al., 2004*), using a multiplex PCR to investigate the presence of five virulence genes in VRE*fm* isolates. All primer sequences are shown in (Table 1).

## ERIC-PCR typing

All isolates were fingerprinted using primers ERIC1A ('5-ATGTAAGCTCCTGGGGATTCAC-3') and ERIC2 ('5-AAGTAAGTGACTGGGGTGAGCG-3') according to a published protocol (*Aljindan, Alsamman & Elhadi, 2018*). ERIC-PCR products were resolved by gel electrophoresis and analyzed on 2% w/v agarose gel with ethidium bromide gel stain in TBE 1 × electrophoresis buffer. A GelPilot 1 kb Plus Ladder (100) (Qiagen, Hilden, Germany) was included as a molecular weight marker. PCR products were visualized using a UV-transilluminator.

## ERIC-PCR data analysis

The clonal relatedness between the strains of *E. faecium* was analyzed using ERIC-PCR fingerprinting with the CLIQS 1D PRO software (TotalLab Ltd., Newcastle, United Kingdom) (*TotalLab, 2015*), Similarity distances between ERIC-PCR profiles were

**Table 1** Oligonucleotide primers for targeted amplification of virulence gene and ERIC sequence.

| Primer name | Sequence (5′–3′) | PCR product size (bp) | Reference |
| --- | --- | --- | --- |
| Primers for amplification of virulence genes of *Enterococcus faecium* | | | |
| aggregation substance (*asa1*) | F: GCACGCTATTACGAACTATGA | 375 | |
| | R: TAAGAAAGAACATCACCACGA | | |
| Gelatinase (*gelE*) | F: TATGACAATGCTTTTTGGGAT | 213 | *Vankerckhoven et al. (2004)* |
| | R: AGATGCACCCGAAATAATATA | | |
| Cytolysin (*cylA*) | F: ACTCGGGGATTGATAGGC | 688 | |
| | R: GCTGCTAAAGCTGCGCTT | | |
| Enterococcal surface protein (*Esp*) | F: AGATTTCATCTTTGATTCTTGG | 510 | |
| | R: AATTGATTCTTTAGCATCTGG | | |
| Hyaluronidase (*Hyl*) | F: ACAGAAGAGCTGCAGGAAATG | 278 | |
| | R: GACTGACGTCCAAGTTTCCAA | | |
| Primer for Amplification of ERIC sequences | | | |
| ERIC1 | F: ATGTAAGCTCCTGGGGATTCAC | 100–10,000 | *Aljindan, Alsamman & Elhadi (2018)* |
| ERIC2 | R: AAGTAAGTGACTGGGGTGAGCG | | |

calculated using the Dice coefficient and dendrograms were constructed based on the unweighted pair group method with arithmetic mean (UPGMA). Normalization steps were included in the analysis of DNA polymorphism patterns produced by ERIC-PCR fingerprinting to ensure an adequate gel-to-gel banding pattern comparison. Isolates with an 80% level of similarity were grouped in the same cluster and were considered clonally related.

# RESULTS

## Sociodemographic distributions

The presentation of the demographic distribution and clinical characteristics of 108 patients is delineated in (Table 2). The age distribution of patients ranged from 1 to 82 years, with a median age of 41.5 years. A higher incidence of *E. faecium* colonization (26.9%; 29/108) was observed in individuals aged between 35 and 49 years. Moreover, the majority of the cohort was comprised of males (55.6%) and approximately (44.4%) were females. Furthermore, it was noted that 56.5% of the patients held a university education, 67.6% were from urban residences, and 28.7% were students by occupation. A high frequency of *E. faecium* was observed in Academy Charity Teaching Hospital (30.6%). In relation to clinical characteristics, the majority of the participants exhibited at least one instance of chronic comorbidity. Gastrointestinal tract infection was the most frequently encountered comorbidity, with 28 participants (25.9%) presenting with this condition. Table 2 reveals that, in accordance with their distribution across wards, the prevalence of enterococci was highest amongst patients who received treatment in internal medicine wards, with 17 individuals (15.7%) identified as carriers. A sizeable proportion of patients (33.3%) had a stay duration of less than one week. The majority of patients (67.6%) reported

self-medication with antibiotics. Non-adherence to prescribed antibiotics was prevalent among an overwhelming majority of patients (74.1%). Additionally, a high proportion (64.8%) of patients had a history of previous antibiotic intake within the preceding three months. In the study, a total of 80 individuals exhibited antibiotic non-adherence behaviors. Among these participants, 45 (56.3%), 65 (81.3%), 52 (65%), and 50 (62.5%) were identified as male, engaging in self-medication, residing in urban areas, and having attained university education, respectively. Additionally, 44 (55%) of the participants demonstrated inadequate knowledge regarding appropriate antibiotic use.

According to the findings reported in (Table 2), a significant proportion of the colonized patients (67.6%) were classified as multidrug-resistant (MDR), while a notable subset of 22 patients (20.4%) exhibited characteristics of extensively drug-resistant (XDR) infections. Approximately 37% of patients were administered antibiotics through a prescription from a physician and 56.5% of individuals under medical care utilized antibiotics as per the recommendations relayed by the pharmacist's prescription. Conversely, a majority of 71 individuals, accounting for 65.7% of patients, ascertained through the study questionnaire that they sought counsel from acquaintances or family members concerning the application of antibiotics. Only 26 (24.1%) of study participants visited the clinic for follow-up after receiving antibiotics prescriptions. According to the findings, a significant proportion of patients, specifically 76 (70.4%), made requests for antibiotics when presenting with flu-like symptoms. Moreover, over 50% of individuals with *E. faecium* infection exhibited lacked knowledge about antibiotic resistance as presented in (Table 2). Based on the results of the chi-square test, there is no statistically significant relationship observed between sociodemographic factors and the occurrence of *E. faecium*. The colonization of *E. faecium* was observed to occur exclusively in subjects without prior history of antibiotic administration, as indicated by a statistically significant $P$-value of $\leq 0.005$.

## Prevalence of antimicrobial resistance among *E. faecium* strains

A total of 108 *E. faecium* isolates were subjected to analysis, with 33 (30.6%) retrieved from Academy Charity Teaching Hospital (ACTH), 29 (26.9%) obtained from Dar-Alelaj Specialized Hospital (DASH), 27 (25%) sourced from Ibrahim Malik Teaching Hospital (IMTH), and 19 (17.6%) originating from Yastabshiroon Hospital Riyadh (YASH). Among them, forty-two strains of colonized *E. faecium* were isolated from the hospitalized patient, while Sixty-six strains were isolated from the non-hospitalized patient (Fig. 1). Out of the total samples collected, 18 (16.7%) were found to be VRE*fm* strains. In our previous work, the majority of *E. faecium* strains have been demonstrated to exhibit resistance to antibiotics with the exception of a single strain that displays susceptibility to all known classes of antibiotics. However, higher and lower resistance rates in *E. faecium* stains were shown against ceftazidime (79.6%), and daptomycin (5.6%) respectively (*Siddig et al., 2022*).

Table 3 reveals that a majority of strains, totaling 73 (67.6%), exhibited MDR, while 22 (20.4%) XDR. None of the identified isolates demonstrated pandrug-resistance (PDR). In general, the prevalence of multidrug-resistant (MDR) *E. faecium* isolates was observed to be highest among non-hospitalized patients, with a frequency of 48/100 (48%), in contrast

**Table 2** Sociodemographic and clinical characteristics of *Enterococcus faecium* isolates among the Khartoum locality hospital patients, Sudan (2018–2020).

| Demographic Characteristics | *Enterococcus faecium* (n = 108) no (%) | Other Enterococcus (n = 62) no (%) | Frequency (n = 170) no (%) | Chi-square Test |
|---|---|---|---|---|
| Gender | | | | $X2 = 0.086, P = 0.768^*$ |
| Male | 60 (55.6) | 33 (53.2) | 93 (22.9) | |
| Female | 48 (44.4) | 29 (46.8) | 77 (45.3) | |
| Age (Years) Mean ± sd | | | | |
| Age groups | | | | $X2 = 3.930, P = 0.415^*$ |
| Less than 20 years | 25 (23.1) | 7 (11.3) | 32 (18.8) | |
| 20–34 years | 28 (25.9) | 20 (32.3) | 48 (28.2) | |
| 35–49 years | 29 (26.9) | 20 (32.3) | 49 (28.8) | |
| 50–64 years | 18 (16.7) | 10 (16.1) | 28 (16.5) | |
| 65 years and above | 8 (7.4) | 5 (8.1) | 13 (7.6) | |
| Educational status | | | | $X2 = 2.401, P = 0.662^*$ |
| Illiterate | 6 (5.6) | 5 (8.1) | 11 (6.5) | |
| Under school age | 5 (4.6) | 5 (8.1) | 10 (5.9) | |
| Primary | 13 (12) | 6 (9.7) | 19 (11.2) | |
| Secondary | 23 (21.3) | 9 (14.5) | 32 (18.8) | |
| University | 61 (56.5) | 37 (59.7) | 98 (57.6) | |
| Residence | | | | $X2 = 1.080, P = 0.298^*$ |
| Urban | 73 (67.6) | 37 (59.7) | 110 (64.7) | |
| Rural | 35 (32.4) | 25 (40.3) | 60 (35.3) | |
| Occupation | | | | $X2 = 4.91, P = 0.672^*$ |
| Employed | 29 (26.9) | 15 (24.2) | 44 (25.9) | |
| Under age | 11 (10.2) | 2 (3.2) | 13 (7.6) | |
| Freelancer | 11 (10.2) | 7 (11.3) | 18 (10.6) | |
| Farmer | 6 (5.6) | 3 (4.8) | 9 (5.3) | |
| Student | 31 (28.7) | 18 (29) | 49 (28.8) | |
| Housewife | 11 (10.2) | 7 (11.3) | 18 (10.6) | |
| Merchant | 6 (5.6) | 7 (11.3) | 13 (7.6) | |
| Retired | 3 (2.8) | 3 (4.8) | 6 (3.5) | |
| Hospital Code | | | | $X2 = 0.377, P = 0.944^*$ |
| ACTH | 33 (30.6) | 20 (32.3) | 53 (31.1) | |
| DASH | 29 (26.9) | 18 (29) | 47 (27.6) | |
| IMTH | 27 (25) | 13 (21) | 40 (23.5) | |
| YASH | 19 (17.6) | 11 (17.7) | 30 (17.6) | |
| Comorbidities | | | | $X2 = 0.923, P = 0.336^*$ |
| Yes | 67 (62) | 43 (69.4) | 110 (64.7) | |
| No | 41 (38) | 19 (30.6) | 60 (35.3) | |

**Table 2** (*continued*)

| Demographic Characteristics | *Enterococcus faecium* (*n* = 108) no (%) | Other Enterococcus (*n* = 62) no (%) | Frequency (*n* = 170) no (%) | Chi-square Test |
|---|---|---|---|---|
| Comorbidities if Yes | | | | |
| Gastrointestinal tract infection | 28 (25.9 ) | 14 (22.6) | 42 (24.7) | |
| Renal and kidney-associated disease | 10 (9.3) | 7 (11.3 ) | 17 (10) | |
| Urinary tract infection | 7 (6.5 ) | 7 (11.3) | 14 (8.2) | |
| Cardiovascular Disease | 2 (1.9 ) | 7 (11.3) | 9 (5.3) | |
| Respiratory tract infection | 8 (7.4 ) | 4 (6.5) | 12 (7.1) | |
| Diabetes | 7 (6.5 ) | 2 (3.2) | 9 (5.3) | |
| Prostatitis | 4 (3.7) | 1 (1.6) | 5 (2.9) | |
| Wound infection | 1 (0.9 ) | 1 (1.6) | 2 (1.2) | |
| Wards (*n* = 70) | | | | $X2 = 1.676, P = 0.642^*$ |
| Surgery | 13 (12 ) | 10 (16.1 ) | 23 (13.5) | |
| Medical | 17 (15.7) | 13 (21 ) | 30 (17.6) | |
| Pediatric | 5 (4.6 ) | 1 (1.6) | 6 (3.5) | |
| ICU | 7 (6.5 ) | 4 (6.5 ) | 11 (6.5) | |
| Patient setting | | | | $X2 = 0.639, P = 0.423^*$ |
| Hospitalized patient | 42 (38.9 ) | 28 (45.2) | 70 (41.2) | |
| Community patients | 66 (61.1) | 34 (54.8 ) | 100 (58.8) | |
| Duration of stay (Days) Mean ± sd | (5 ± 2) | | | |
| Duration of stay (*n* = 70) | | | | $X2 = 0.603, P = 0.437^*$ |
| Less than week | 36 (33.3 ) | 22 (35.5 ) | 58 (82.9) | |
| Week and more | 6 (5.6 ) | 6 (9.7 ) | 12 (17.1) | |
| Self-Medication | | | | $X2 = 1.854, P = 0.173^*$ |
| Yes | 73 (67.6 ) | 48 (77.4 ) | 121 (71.2) | |
| No | 35 (32.4) | 14 (22.9 ) | 49 (28.8) | |
| Antibiotic adherence | | | | $X2 = 1.492, P = 0.221^*$ |
| High Adherence | 28 (25.9 ) | 11 (17.7 ) | 39 (22.9) | |
| Non-adherence | 80 (74.1 ) | 51 (82.3) | 131 (77.1) | |
| Used Antibiotic in last 3 months | | | | $X2 = 0.212, P = 0.645^*$ |
| Yes | 70 (64.8) | 38 (61.3) | 108 (63.5) | |
| No | 38 (35.2) | 24 (38.7 ) | 62 (36.5) | |
| Antibiotic exposure | | | | $X2 = 29.55, P = 0.005^{**}$ |
| Ceftazidime | 17 (15.7) | 3 (4.8 ) | 20 (11.8) | |
| Ceftriaxone | 13 (12 ) | 4 (6.5 ) | 17 (10) | |
| Clindamycin | 1 (0.9 ) | 4 (6.5 ) | 5 (2.9) | |
| Amoxicillin | 5 (4.6 ) | 8 (12.9) | 13 (7.6) | |
| Gentamicin | 24 (22.2) | 4 (6.5 ) | 28 (16.5) | |
| Ciprofloxacin | 4 (3.7 ) | 6 (9.7 ) | 10 (5.9) | |
| Azithromycin | 2 (1.9 ) | 3 (4.8 ) | 5 (2.9) | |
| Chloramphenicol | 2 (1.9 ) | 1 (1.6 ) | 3 (1.8) | |
| Erythromycin | 5 (4.6 ) | 2 (3.2) | 7 (4.1) | |

**Table 2** (*continued*)

| Demographic Characteristics | *Enterococcus faecium* (*n* = 108) no (%) | Other Enterococcus (*n* = 62) no (%) | Frequency (*n* = 170) no (%) | Chi-square Test |
|---|---|---|---|---|
| Metronidazole | 3 (2.8) | 2 (3.2) | 5 (2.9) | |
| Penicillin | 2 (1.9 ) | 4 (6.5) | 6 (3.5) | |
| Tetracycline | 9 (8.3 ) | 3 (4.8 ) | 12 (7.1) | |
| Vancomycin | 2 (1.9) | 5 (8.1 ) | 7 (4.1) | |
| Not remembered | 19 (17.6 ) | 13 (21) | 32 (18.8) | |
| Antimicrobial categorize | | | | $X2 = 3.560, P = 0.059^*$ |
| MDR | 73 (67.6) | 44 (71 ) | 117 (68.8) | |
| XDR | 22 (20.4 ) | 5 (8.1) | 27 (15.9) | |
| Taking antibiotics according to physician consultation | | | | $X2 = 0.710, P = 0.399^*$ |
| Yes | 40 (37 ) | 19 (30.6 ) | 59 (34.7) | |
| No | 68 (63) | 43 (69.4 ) | 111 (65.3) | |
| Taking antibiotics according to pharmacist consultation? | | | | $X2 = 0.164, P = 0.684^*$ |
| Yes | 61 (56.5) | 37 (59.7) | 98 (57.6) | |
| No | 47 (43.5) | 25 (40.3) | 72 (42.4) | |
| Taking antibiotics according to friends or relative consultation? | | | | $X2 = 0.491, P = 0.483^*$ |
| Yes | 71 (65.7) | 44 (71) | 115 (67.6) | |
| No | 37 (34.3) | 18 (29) | 55 (32.4) | |
| Would you visit a physician for a follow-up after taking antibiotics? | | | | $X2 = 0.504, P = 0.477^*$ |
| Yes | 26 (24.1 ) | 18 (29) | 44 (25.9) | |
| No | 82 (75.9) | 44 (71 ) | 126 (74.1) | |
| If ill with flu-like symptoms and the doctor doesn't prescribe antibiotics, do you take an antibiotic? | | | | $X2 = 0.284, P = 0.594^*$ |
| Yes | 76 (70.4) | 46 (74.2 ) | 122 (71.8) | |
| No | 32 (29.6 ) | 16 (25.8 ) | 48 (28.2) | |
| Do you aware of miss use of antibiotics leads to resistance of bacteria? | | | | $X2 = 2.538, P = 0.111^*$ |
| Yes | 59 (54.6) | 26 (41.9) | 85 (50) | |
| No | 49 (45.4 ) | 36 (58.1) | 85 (50) | |

**Notes.**
Academy Charity Teaching Hospital (ACTH); Dar-Alelaj Specialized Hospital (DASH); Ibrahim Malik Teaching Hospital (IMTH); Yastabshiroon Hospital Riyadh (YASH); Multidrug-resistant (MDR); and extensively drug-resistant (XDR). **$P < 0.05$ is significant.

to a rate of 25/70 (35.7%) among hospitalized patients. Conversely, the occurrence of extensively drug-resistant (XDR) isolates was equal among the hospitalized and non-hospitalized patients, with a rate of 13%. The results of a chi-square analysis revealed a statistically significant correlation between the categorization of antimicrobial agents and community-based patients ($P = 0.021$), as evidenced by the data presented in (Table 3).

## Virulence genes among VRE*fm* strains

Among 108 *E. faecium* isolates, a total of 11 VRE*fm* strains tested positive for virulence genes as shown in (Table 4). The present study reveals that the most common virulence-encoding
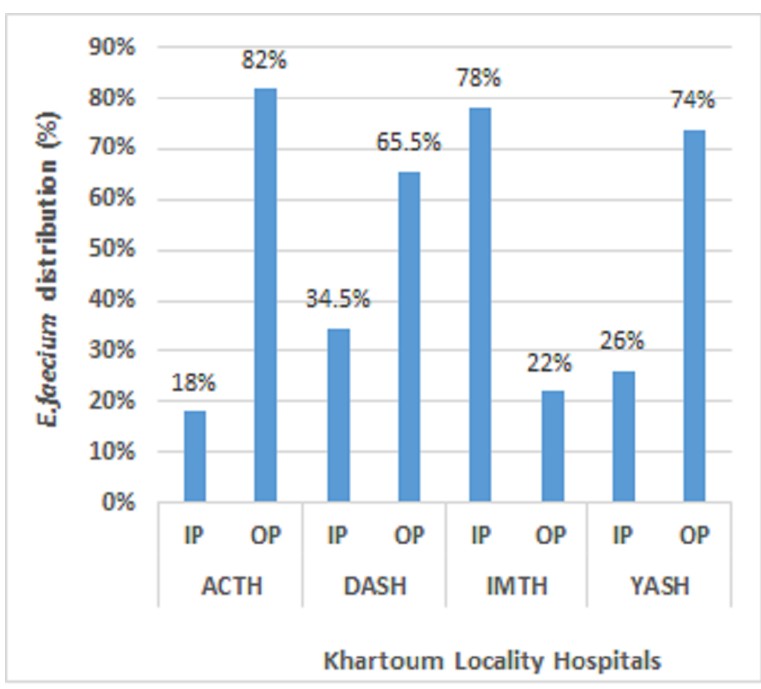

**Figure 1 Distribution of *E. faecium* among the patients at Khartoum locality hospital, Sudan, 2018–2020.** Hospitalized patients (IP); Non-hospitalized patients (OP); Academy Charity Teaching Hospital (ACTH); Dar-Alelaj Specialized Hospital (DASH); Ibrahim Malik Teaching Hospital (IMTH); and Yastabshiroon Hospital Riyadh (YASH) (*Siddig et al., 2022*).

**Table 3 Antimicrobial categorization of *E.faecium* isolates among the hospitalized and non-hospitalized patients.**

| Antimicrobial categorize | Hospitalized (N = 70) | | P | Non-hospitalized (N = 100) | | P |
|---|---|---|---|---|---|---|
| | *E. faecium* | Other Enterococcus | | *E. faecium* | Other Enterococcus | |
| Susceptible | 8 (11.4) | 6 (8.6) | 0.751 | 5 (5) | 7 (7) | 0.021 |
| MDR | 25 (35.7) | 18 (25.7) | | 48 (48) | 26 (26) | |
| XDR | 9 (12.9) | 4 (5.7) | | 13 (13) | 1 (1) | |

genes among VRE*fm* were *asa*1 4 (22.2%), followed by *esp* 3 (16.7%), *hyl* 1 (5.6%), *gel*E 1 (5.6%), and *gel*E-*hyl* 2 (11.1%). Notably, no instance of the *cyl*A gene was discerned within the studied population.

### ERIC-PCR analysis of *E. faecium*

The present study employed the ERIC-PCR DNA fingerprinting technique on a total of 108 *E. faecium*. Subsequent analysis revealed ERIC profiles spanning a range of 4-13 distinct banding patterns, with molecular weights ranging from 100 to 5,000 bp (Fig. 2). The assessment of ERIC fingerprinting patterns utilizing the Dice coefficient and UPGMA presents that ERIC-PCR profiles demonstrate considerable genetic diversity among the isolated samples. The rate of *E. faecium* isolates was found to be 13% (14/108) when clustering was based on a 100% similarity coefficient. Additionally, 52.8% (57/108) of the

**Table 4 Virulence gene patterns among the VRE*fm* isolates.**

| Species | Virulence factors | Number of positive isolates |
| --- | --- | --- |
| | *asa1* | 4 (22.2%) |
| | *hyl* | 1 (5.6%) |
| VRE*fm* (*n* = 18) | *gelE* | 1 (5.6%) |
| | *esp* | 3 (16.7%) |
| | *gelE-hyl* | 2 (11.1%) |

**Notes.**
Enterococcal aggregation substance (asa1); hyaluronidase (hyl); gelatinase (gelE); Enterococcal surface protein (esp).

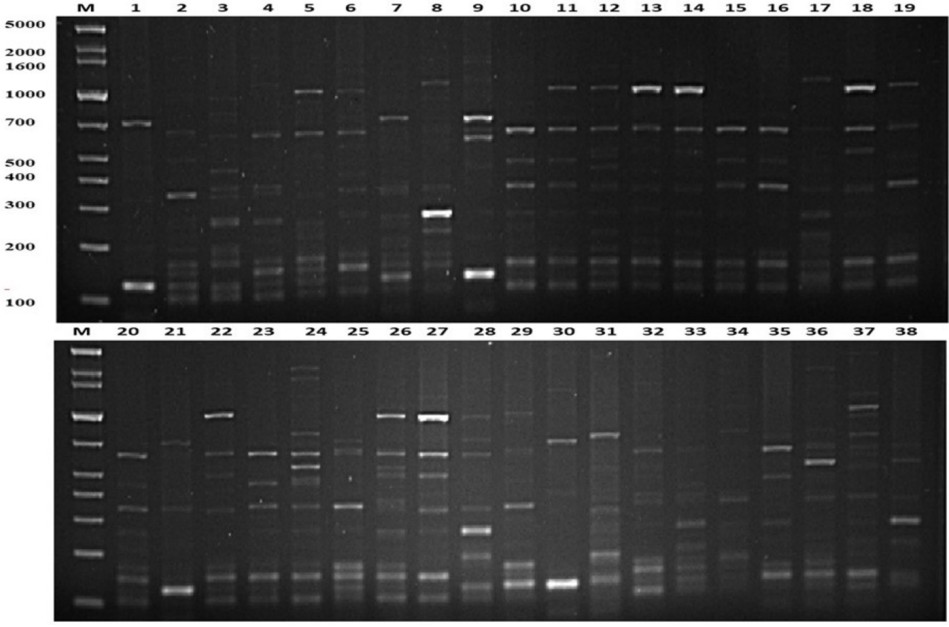

**Figure 2 Representative genetic profiles yielded by the ERIC-PCR analysis of *E.faecium* strains isolated from different Khartoum locality hospitals.** Lanes M = Gel Pilot 1 Kb Plus ladder; lanes 1–19, 20–38 = A representation of the ERIC profiles for *E. faecium* isolates.

isolates were found to cluster with an 80% similarity coefficient. Whereas 24.1% (26/108) isolates of *E. faecium* had less than 80% similarity. Furthermore, 10 strains demonstrating a singular lineage below 50% were omitted, and a grouping analysis revealed that fourteen patients with identical fingerprint patterns were allocated to clusters A through G, as depicted in (Fig. 3).

## DISCUSSION

Until the 1980s, *Enterococcus* spp. were merely intestinal microbes of little clinical significance. Now, they are among the most common nosocomial pathogens, so physicians are becoming more worried (*Arias & Murray, 2008*). Resistance in enterococci has increased dramatically and the incidence of VRE colonization spread where vancomycin is one of the

Peerj

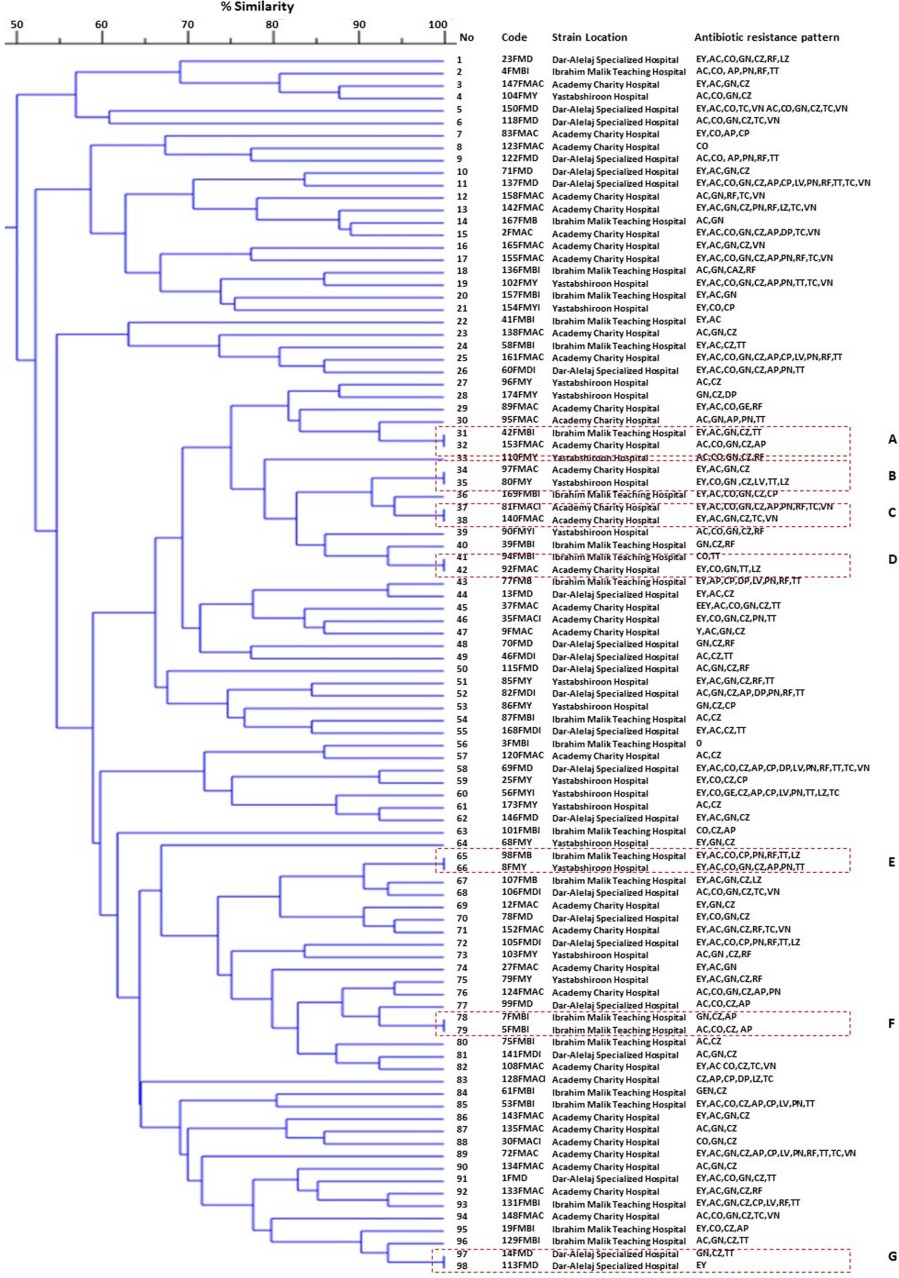

**Figure 3** **A represented dendrogram of ERIC-PCR using CLIQS fingerprint data software and UPGMA with arithmetic averages at 80% similarity on 108 strains of *Enterococcus faecium* isolated from Khartoum locality hospitals.** The different clusters at 80% similarity are arbitrarily designated as sub-group 1–53. In the code, the first number refers to the patient, the second letter refers to the strain, and the third refers to admitted hospital's name. ERIC-PCR strains of *E. faecium* were investigated in this study into seven groups, every strain with 100% genetic similarity represented in clusters (A–G).

antibiotics of choice used to treat infections caused by Gram-positive multidrug-resistant organisms, such as Enterococci. In Sudan, no reports are available on the molecular epidemiology analysis of *E. faecium* and there are limited data on the prevalence of antibiotic resistance profiles. We previously conducted a study on ''Prevalence and Antimicrobial Profile of Colonized Enterococcus Species Isolated from hospitalized and non-hospitalized Patients, Khartoum, Sudan'' (*Siddig et al., 2022*), but the study did not address risk factors of acquisition of *E. faecium* infection associated with Sociodemographic characteristics. In this study, antibiotic resistance of *E. faecium* carrier rates, sociodemographic and clinical characteristics associated with *E. faecium* colonization were studied. VRE*fm* virulence genes and the genetic relationship among *E. faecium* strains isolated from Khartoum locality hospitals were investigated.

In the present study, the prevalence of *E. faecium* strains was 63.5% (108/170). Our findings showed that a high prevalence of *E. faecium* occurred in non-hospitalized patients of Yastabshirron Hospital Riyadh (70%), whereas, *E. faecium* occured in 67.7% of hospitalized patients of Ibrahim Malik Teaching Hospital. These results were consistent with a previous study done by *Tedim et al. (2015)* revealing that the prevalence of colonization with *E. faecium* is higher in non-hospitalized patients compared to hospitalized patients. Antimicrobial resistance percentages among bacteria from human illnesses in the European Union and European Economic Area countries (EU/EEA) did not vary significantly between 2014 and 2020, except for *E. faecium*, where the proportion of vancomycin resistance increased from 9% in 2014 to 17% in 2020 (*European Centre for Disease Prevention and Control, 2022*), and this consists with our result where the prevalence of VRE*fm* was (16.6%).

Broad-spectrum antibiotics have the potential to harm the normal anaerobic flora of the gastrointestinal tract, resulting in infectious diseases due to their bactericidal impact against Enterococcus. Many studies have reported that previous use of broad-spectrum antibiotics is a risk factor for acquiring multidrug-resistant pathogens (*Tenney et al., 2018*; *Son et al., 2021*). However, few investigations have been undertaken to evaluate the link between previous antibiotic exposure and the acquisition of *E. faecium* strains (*Son et al., 2021*). Our findings indicated a significant association between prior antibiotic exposure and the acquisition of *E. faecium* infection ($P \leq 0.005$). Interestingly, our study data findings supported that overusing antibiotics increased the likelihood of resistance while decreasing their efficacy, this was demonstrated by the high prevalence of antibiotic usage among hospitalized patients and non-hospitalized patients, as well as by their use of self-medication and erratic antibiotic regimen.

In several reported cases, gastrointestinal tract colonization generally precedes infection with antibiotic-resistant *E. faecium*, in particular, intestinal overgrowth by antibiotic-resistant enterococci is a recognized risk factor for disease (*Montealegre, Singh & Murray, 2016*; *Banla, Salzman & Kristich, 2019*). Our study showed the prevalence of *E. faecium* colonization was higher among patients who had gastrointestinal tract infections (25.9%) compared to other chronic infections. The high prevalence rate of self-medication and antibiotic usage seen in this study could partly be explained by the patient's desire for a fast

recovery from the disease. The economic situation is another major cause for self-medicates, and consultations of friends or relatives to avoid paying the physicians' fees.

A recent study conducted in primary healthcare centers in Qatar reported that many factors contribute to the increased incidence of bacterial resistance to antibiotics, particularly; the misuse of antibiotics by physicians and the easy acquisition of antibiotics *via* non-physicians (*Alkhuzaei et al., 2018*). In the developing world, it is common practice for pharmacies to distribute antibiotics based on patient requests, and this study is in line with our results that showed 56.5% of patients have taken antibiotics according to pharmacist consultation. The findings of the study also reveal more than half of patients received antibiotics based on recommendations provided by friends or relatives. In addition, even in cases when they weren't necessary, such as flu-like symptoms, 76 (70.4%) of patients demanded antibiotics. This understanding was explained by their knowledge which led them to believe that antibiotics were helpful in such circumstances (*WHO, 2021*). According to a study conducted among community pharmacies in Addis Ababa, an increase in the frequency of over-the-counter sales of antibiotics has been observed. These results are attributable to commercial interests, consumer pressure, and lax rules (*Gebretekle & Serbessa, 2016*).

Proper adherence to antibiotic therapy is an essential element among various measures necessary to combat the rising issue of antimicrobial resistance (*Endashaw Hareru et al., 2022*). In this paper, we describe this burden and its consequential effects on medication adherence in patients at Khartoum locality hospitals. We assigned everyone who was not perfectly adherent to the use of antibiotics as "Non-adherence". The overall prevalence of drug non-adherence was found to be 74%. The study findings are consistent with studies done in the USA (70%) (*Kuo, Haftek & Lai, 2017*), Ethiopia (60.1%) (*Endashaw Hareru et al., 2022*), and Nigeria (63.4%) (*Kehinde & Ogunnowo, 2013*). Our findings are also higher than previous studies conducted in Malaysia (34.8%) (*Ab Halim et al., 2018*).

A multiplex PCR developed for the simultaneous detection of *E. faecium* virulence genes that encode for aggregation substance (*asa* 1), gelatinase (*gel* E), cytolysin (*cyl* A), enterococcal surface protein (*esp*), and hyaluronidase (*hyl*). Our result showed that virulence genes *asa* 1 (22.2%), followed by *esp* gene (16.7%) are predominant in the virulent patterns of VRE*fm* isolated from hospitals and communities. Findings from our current study are consistent with a recent study from Southern California and Puerto Rico was reported the *asa* 1 gene is predominant in enterococci isolated from hospitals, the natural environment, animals, and wastewater (*Ferguson et al., 2016*).

According to the dendrogram, strains with 100% similar ERIC profiles were found in clusters, A (31/32), B (34/35), D (41/42), and cluster E (65/66) were isolated from patients among different hospital wards and communities. On the other hand, isolates presented in clusters, C (37/38), F (78 /79), and G (97/98) with similar ERIC profiles isolated from patients within the same hospitals. *E. faecium* strains show high genetic diversity among isolates raising the possibility of circulation of various *E. faecium* strains between the hospitals and the community.

## CONCLUSIONS

Our current study investigates the prevalence rate of *E. faecium* antibiotic resistance, sociodemographic characteristics, virulence genes, and the genetic relationship of *E. faecium* isolated from hospitalized and non-hospitalized patients from localized hospitals in Khartoum. Acquisition of *E. faecium* infection with the most supporting data showed that the previous history of antibiotic usage played a role as a risk factor. Appropriate antibiotic-resistance testing programs, as well as competent antibiotic stewardship, are critical in successfully lowering resistance to the aforementioned drugs, particularly in VRE isolates. In this study, antibiotic non-adherence was considerably high among the participants. As a result, community service providers must provide relevant prescription information as well as appropriate counseling to antibiotic non-adherent patients. This study also showed that there is an urgent need for education programs targeting all levels of the community and directed toward changing the public attitude and behavior to rationalize antibiotic use and limit self-medication and overuse. Furthermore, strict policies must be enforced to regulate the procurement of antibiotics and prohibit their purchase without a prescription. Our study also concluded that ERIC-PCR is a reliable typing method for discriminating different isolates of *E. faecium* isolated from hospitalized and non-hospitalized patients.

### Funding
The authors received no funding for this work.

### Competing Interests
The authors declare there are no competing interests.

### Author Contributions
- Loai Abdelati Siddig conceived and designed the experiments, performed the experiments, analyzed the data, prepared figures and/or tables, authored or reviewed drafts of the article, and approved the final draft.
- Magdi Bayoumi conceived and designed the experiments, performed the experiments, analyzed the data, prepared figures and/or tables, authored or reviewed drafts of the article, and approved the final draft.
- Nasreldin Elhadi conceived and designed the experiments, performed the experiments, analyzed the data, prepared figures and/or tables, authored or reviewed drafts of the article, and approved the final draft.

### Human Ethics
The following information was supplied relating to ethical approvals (*i.e.*, approving body and any reference numbers):

Ethical approval for this study was obtained from the Graduate College-UMST and the Federal Ministry of Health, Sudan (ethical application Ref: ws-wk-aete-44-a).

## Data Availability

The raw data are available in the Supplemental File.

## Supplemental Information

Supplemental information for this article can be found online at http://dx.doi.org/10.7717/peerj.16169#supplemental-information.

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
