# Peer review of "Sociodemographic distributions and molecular characterization of colonized Enterococcus faecium isolates from locality hospitals in Khartoum, Sudan"

_PeerJ, doi:10.7717/peerj.16169_

## Round 0.1 · original submission · Major Revisions

Please rewrite the manuscript according to reviewers' comments and highlight the changes.

·

Basic reporting

The introduction, aim and objectives of the study is clearly written and well understood. It contains sufficient literature and relevant to the subject matter.
Raw data are well represented and the right statistical tools were used to analyse the data.
However, the Figure 1 is quite clear. See my comments below:
COMMENTS:
1. The figure is not a properly labelled e.g. X and Y axis should be well labelled
2. It is not a true reflection of the Figure title i.e. the % of hospitalized and non-hospitalized should be well displayed side by side in the bars
3. Data labels should be shown on all bars or delete data labels
4. Possibly delete grid lines
5. Show axis lines e.g. Y-axis and X-axis lines
6. Show key for abbreviated antibiotics on the X-axis

Experimental design

No comment

Validity of the findings

No comment

Additional comments

A very good manuscript, however, a summary of strain location and antibiotic resistance pattern should have been analyzed to identify the distribution pattern of the resistant strains among the hospitals

Reviewer 2 ·

Basic reporting

A. Improvements in language is needed for clarity, detailed as the following:
1. Please break the long sentences for better understanding and removal of grammar issues. Examples are the sentences in line 22-26 and 253-258.
2. Please provide definition for the terms such as “extensively drug-resistant” (first appearance line 39) and PDR (line188).
3. Please define the meaning of “low adherence” (line 163) and “residual inhibitory impact” (line 231) for better understanding.
4. Italic “E. faecium” in line 52, “Enterococcus faecium” in line 53, “Enterococcus” in line 69. Change “Enterococci” in line 58 and “enterococcus” in line 79 to “Enterococcus”. Full bacterial name is only needed for the first appearance, thus change “Enterococcus faecium” to “E. faecium” in line 66, 228, and 286.
5. Please correct grammar issues throughout this paper.

B. Please provide more introductory information about the ERIC-PCR typing method. For example, what are the ERIC sequences? How prevalent is ERIC in Enterococcus faecium? Any previous application of ERIC-PCR in Enterococcus?

C. Data presentation:
1. Fig 1 – axis title is needed.
2. Fig 2 – what is “[i]”?
3. Fig 3 – what’s the meaning of A-G?

Experimental design

Lack of details in experimental procedures prevents effective evaluation of the rigorousness of this paper.
Specifically, in line 109-111, please provide detailed and complete description for the conducted phenotypic tests and selection criteria for the test results. This info is necessary to validate the strain identification process.

Validity of the findings

Repetitive presentation of previously published data is seen.
Specifically, in line 182-186, 230-232, and Fig 1, it seems that the discussed data have already been published in ref 5, in which these data were presented as tables. Please edit the paper through to make clear differentiation between published and new data and provide appropriate citations to previous work.

---

## Round 0.2 · Minor Revisions

Dear Dr. Siddig and Dr. Elhadi,

Please address the comments and make the required minor revisions.

Reviewer 2 ·

Basic reporting

Majority of the previous comments on basic reporting have been addressed. Some minor modifications are still needed:

Table 4: put the title above the table.

Fig 1: Abbreviations for antibiotics are not needed in the figure legend when full antibiotic names are provided in the figure.

Fig 2: In the figure legend, 1) define “M”; 2) it looks like lane 1 and 21 are also samples, as the lanes for the ladder were labeled as “M”, please correct; 3) delete “[i]”, or change it to the correct citation if it was used to indicate a need for reference.

Fig 3: In the figure legend, define A-G.

Experimental design

Details of the phenotypic tests and selection criteria have been added to the method section. Previous comment was addressed.

Validity of the findings

Please see the additional comments

Additional comments

Please address the copyright issue: Figure 1 is a republication of data presented in Table 3 of ref 5 (full text link: http://pubs.sciepub.com/ajidm/10/4/1/index.html). Please provide citation in both main text and figure legend and obtain appropriate republication permission.

---

## Round 0.3 · Minor Revisions

Comments:
1. Throughout the text: E. faecium instead of E.faecium
2. Introduction: Please add 1-2 sentences describing symptoms caused by E. faecium.
3. Discussion: Lines 302, 304:
You write: The overall prevalence of drug non-adherence was found to be 74%. The study findings are consistent with studies done in USA (70%)….Our findings are also lower than previous studies conducted in Malaysia (34.8%).
Please clarify. 74% is higher than 34.8%.

---

## Round 0.4 · accepted · Accept

The authors have addressed all of the reviewers' comments.